# Use of Intrinsic Capacity Domains as a Screening Tool in Public Health

**DOI:** 10.3390/ijerph20054227

**Published:** 2023-02-27

**Authors:** Livia Maria do Nascimento, Thainá Gabriele Camargo da Cruz, Juliana Fernanda de Lima e Silva, Letícia Prado Silva, Beatriz Bigatão Inácio, Carolina Masumi Oki Sadamitsu, Marcos Eduardo Scheicher

**Affiliations:** 1Graduate Program in Human Development and Technologies, Institute of Biosciences, Paulista State University—UNESP, Rio Claro 13506-900, SP, Brazil; 2Department of Physical Therapy and Occupational Therapy, Paulista State University—UNESP, Marília1 7525-900, SP, Brazil

**Keywords:** intrinsic capacity, aging, screening tests

## Abstract

The World Health Organization (WHO) created the concept of Integrated Care for Older People and one of its constructs is intrinsic capacity (IC). The study aimed to carry out a screening with the tools designated by the WHO to assess the IC domains and whether they can be used as indicators for decision-making on integrated care for older people based on risk categorization. The interaction between the risk category and the domain scores was verified. One hundred and sixty three (163) community-dwelling older people of both genders were evaluated. Domains assessed: cognitive, psychological, vitality, locomotion, and sensory. Scores indicating a low, moderate and high risk were assigned to each domain. For all domains, there were individuals in all risk groups. Effect of risk on the domains: cognitive [χ^2^(2) = 134.042; *p* < 0.001], psychological [χ^2^(2) = 92.865; *p* < 0.001], vitality [χ^2^(2) = 129.564; *p* < 0.001], locomotion [χ^2^(2) = 144.101; *p* < 0.001], and sensory [χ^2^(2) = 129.037; *p* < 0.001]. Scores of the CI domains were affected by the risk category. There were individuals in all risk groups, demonstrating the importance of screening as a public health strategy, making it possible to know which risk category each elderly person belongs to and thus develop strategies in the short-, medium- and long-term.

## 1. Introduction

Population aging is a worldwide phenomenon. The number of people over the age of 60 years is estimated to reach 2 billion by 2050 [1]. In addition to excessive cellular and molecular modifications, aging is caused by an accumulation of damage, which leads to a gradual loss of physiological reserve that results in greater vulnerability, a higher risk of developing diseases, and greater functional dependence [1]. In view of the growth of the elderly population, the World Health Organization (WHO) created the concept of Integrated Care for Older People (ICOPE) that arose from the need to overcome traditional health models focusing only on the disease, deficits, and limitations. One of the constructs used by this strategy is intrinsic capacity (IC), which is defined as the composite of all the mental and physical capacities that a person can draw on over a lifetime [2]. Five domains were considered essential but not unique for structuring IC: locomotion (balance, gait, and muscle strength), vitality (balance between energy production and consumption), cognition (memory, intelligence, and problem solving), psychological (mood and sociability), and sensory (hearing and vision) [3]. The assessment of IC domains permits contextualized monitoring considering the specificities of the individual and the population analyzed, as well as to evaluate the effectiveness of already established interventions or to indicate necessary short-, medium- and long-term interventions. Based on this assessment, public health authorities can, for example, identify care needs, monitor critical signs, and avoid care dependency [4,5]. In addition, they can propose care models for the maintenance of domains that were not affected by decline. Step 1 of ICOPE proposes screening people based on the five IC domains, and individuals with IC losses are referred to step 2 on person-centered assessment in primary care [6]. Only four studies evaluated step 1 proposed by the WHO, showing a gap of knowledge [7]. Thus, the objective of this study was to carry out a screening with the tools designated by the WHO to assess the IC domains and whether they can be used as indicators for decision-making on comprehensive health care for the elderly based on risk categorization. In addition, it was verified whether there is an interaction between the risk category and the domain scores.

## 2. Materials and Methods

Individuals aged 60 years or over who lived in the catchment area of the two Family Health Units (USFs in the Portuguese acronym) of the city of Marília, São Paulo, Brazil, participated in the study. A total of 1113 older adults are registered with the two USFs. The Ethics Committee of the Faculty of Philosophy and Sciences, Unesp, Marília, approved the study (Approval number 4.168.934).

Inclusion criteria: aged 60 years or over, both sexes, resident of the area covered by the USFs. Exclusion criteria: not signing the free and informed consent form, after agreeing to participate in the research.

A search of individuals aged 60 years or over who were registered with the USFs was performed through the e-SUS Primary Care digital platform. Next, the assessment was scheduled in two ways: (1) the older adults were contacted by telephone and invited to come to the USF to participate in the study; (2) community agents of the USFs visited the homes of the older adults and performed the assessments on those who agreed to participate.

### 2.1. Assessment of Intrinsic Capacity Domains

#### 2.1.1. Cognitive Domain

The Montreal Cognitive Assessment (MoCA) was used to assess the cognitive domain of the participants. The total possible score of the test is 30 points and scores are classified as follows: ≥25 points: low risk/normal; 21 to 24 points: moderate risk/mild cognitive impairment (MCI); <21 points: high risk/Alzheimer’s disease, with sensitivity of 82.2% and specificity of 92.3% [8].

#### 2.1.2. Psychological Domain

The 15-item Geriatric Depression Scale (GDS-15) was used to analyze the psychological domain. Its advantages include easy-to-understand questions, small variation in the possible answers (yes/no), and self-administration or administration by an interviewer. The score ranges from 0 (absence of depressive symptoms) to 15 points (maximum score of depressive symptoms). Almeida and Almeida [9] suggested a cut-off score ≥5 to indicate the presence of depressive symptoms in older adults. Paradela et al. [10] proposed the following cut-off scores: 0 to 5 points: low risk of depression; 6 to 10 points: moderate risk of depression; 11 to 15 points: high risk of depression, with sensitivity of 81% and specificity of 71%.

#### 2.1.3. Sensory Domain

Hearing and visual impairments were assessed by self-report using previously validated questions, which were found to be accurate when compared to objective measures [11,12]. Hearing status was evaluated by asking the participants to rate their hearing as excellent, very good, good, fair, or poor (with the use of a hearing aid, if usually worn), with sensitivity of 66.9% and specificity of 85.1% [12]. For vision assessment, the participants were asked “How good is your vision for seeing things at a distance, for example, recognizing a friend across the street?” and “How good is your vision for seeing close up things, for example, reading a newspaper?”. The response options were categorized as excellent, very good, good, fair, or poor. This assessment was performed with the participant wearing glasses or corrective lenses if they usually did. Self-reported vision obtained a sensitivity of 79% and specificity of 75% [11]. The responses regarding hearing and vision were scored on a Likert-type scale: 1—poor, 2—fair, 3—good, 4—very good, and 5—excellent. The responses were classified as follows: 1 and 2—high risk of problems in the sensory domain; 3—moderate risk, and 4 and 5—low risk.

#### 2.1.4. Vitality Domain

The term vitality is understood as the bodily functions devoted to metabolizing dietary intake in order to produce the amount of energy necessary for maintaining optimal homeostasis [13]. With aging, changes occur in energy expenditure and metabolism that can interfere with food intake [14]. To maintain proper function, the body needs to balance energy intake and expenditure. Since there is no consensus on the best approach to assessing the vitality domain, the Mini-Nutritional Assessment (MNA), a tool recommended by the WHO, was used [15]. According to the WHO, acceptance of the MNA by older adults is good, showing 80% sensitivity and 90% specificity [16]. The instrument was translated and validated for Brazilian Portuguese by Machado et al. [17]. The following scores were considered for screening: 12 to 14 points: low risk; 8 to 11 points: moderate risk; ≤7 points: high risk [18].

#### 2.1.5. Locomotion Domain

The Short Physical Performance Battery (SPPB) was used to assess the locomotion domain. This instrument evaluates lower limb function through a battery of tests of static balance, gait speed, and five-times repeated chair sit-to stand (lower limb strength). The tests can be evaluated individually or combined, generating a total score that ranges from 0 (worst performance) to 12 (best performance) [19]. The following risk criterion was adopted: 0 to 6 points: high risk; 7 to 9 points: moderate risk; 10 to 12 points: low risk [20].

### 2.2. Statistical Analysis

The distribution of the variables (parametric or nonparametric) was verified by the Kolmogorov-Smirnov test. Data were expressed as the mean ± standard deviation and the effect of risk on the IC domains was evaluated by one-way ANOVA, with pairwise comparisons by the Bonferroni post-test. The cognitive domain scores were corrected for educational level. A classification (high, moderate, and low risk of developing a given problem) was assigned according to the cut-off scores of each test described above. Based on this classification, the absolute frequency in each risk group was determined. A *p* value < 0.05 was considered to be significant. The analyses were performed using the SPSS 20.0 software.

## 3. Results

A total of 163 community-dwelling older adults from the city of Marília, São Paulo, Brazil, were evaluated (16.64% of all older adults registered with the USFs). The characteristics of the participants can be seen in Table 1.

A risk interaction was observed in the CI domains: cognitive [χ^2^(2) = 134.042; *p* < 0.001, η^2^: 0.64], psychological [χ^2^(2) = 92.865; *p* < 0.001, η^2^: 0.74], vitality [χ^2^(2) = 129.564; *p* < 0.001, η^2^: 0.86], locomotion [χ^2^(2) = 144.101; *p* < 0.001, η^2^: 0.83], and sensory [χ^2^(2) = 129.037; *p* < 0.001, η^2^: 0.58], indicating the large effect of risk category on domain scores.

Figure 1, Figure 2, Figure 3, Figure 4 and Figure 5 show the comparisons (A) and the absolute distributions (B) between the risk levels for each evaluated domain. It is possible to observe that in all Figures and, therefore, in all domains evaluated and categorized by risk, there was a significant difference in the mean scores in the comparison between low and high risk, making clear the importance of screening to know which risk category the elderly is classified and thus make decisions.

Figure 1, for example, shows a large number of people categorized as high and moderate risk of developing dementia or mild cognitive impairment (B) and how their scores differed significantly from those categorized as low risk (A). In Figure 2, similarly, it is possible to observe a large number of people categorized as high and moderate risk of having self-reported sensory impairments (B) and how the scores differed significantly from people categorized as low risk (A).

In Figure 3, although a small number of elderly people were categorized as high and moderate risk of having psychological disorders (B), there was also a significant difference in relation to the scores between the risk categories (A). Few people categorized as high/moderate risk does not mean they do not require short-term attention.

In Figure 4, there is a large number of elderly people in the high and moderate risk categories of developing locomotor impairments (B) and a significant difference in relation to the scores between the risk categories (A). The presence of people in two categories with a high probability of developing locomotor disorders means that the public authorities need to pay attention to this population.

In Figure 5, a small number of elderly people categorized as high risk for disturbances in the vitality domain can be observed, compared to those at low risk (B). Despite this, there was a significant difference in scores between risk categories (A). Few people classified as high risk does not mean that they do not need care and that it should be short-term, mainly because this domain evaluated the issue of malnutrition.

## 4. Discussion

Population aging is an unprecedented phenomenon. In Brazil, IBGE data project that the elderly population will increase from 9.2% in 2018 to 25.5% of the country’s total population in 2060 [21]. Together with the demographic growth, challenges of this population arise that include comorbidities, geriatric syndromes, and the decrease in physiological reserve [22,23]. In view of this situation, the WHO developed the concept of comprehensive care for the elderly with emphasis on IC in an attempt to change paradigms of elderly health, focusing on prevention and the maintenance of capacities [2]. Particularly within the context of public health, the availability of tools that assist in short-, medium- and long-term decision-making is important.

In 2015, the WHO established healthy aging as that during which the older adult maintains autonomy and independence and defined it as “the process of developing and maintaining the functional ability that enables well-being in older age” [24]. To provide healthy aging, public health authorities must know the profile of the population and establish goals to achieve the objective.

The results of the present study showed an effect of the risk category on the domains and that there were individuals in all risk categories, indicating that this stratification is important. The WHO itself established that the first step in the implementation of ICOPE is the screening of the population [7]. The effect of the degree of risk on the domains means that elderly people categorized as high/moderate risk are much more likely to already have or develop the disorder than those stratified as low risk.

Screening evaluations are aimed at indicating developmental conditions of a given clinical situation. In this respect, a comprehensive assessment of the elderly can identify those who are undergoing the process of physiological aging and need care over time to maintain their skills, as well as those who have an accelerated physiological decline and need immediate care.

In the cognitive domain assessed by the MoCa, 55 older adults (34%) had scores that classified them as having cognitive impairment without dementia or MCI. The MoCa was developed as a screening test to detect MCI [25] and its properties are superior to the Mini-Mental State Examination [26]. In a proportion of older adults, cognition cannot be classified as normal or dementia [27]; cognitive alterations are present while functional abilities are preserved [28]. Studies have shown that individuals with MCI are at increased risk of developing dementia [29,30,31]. In a two-year follow-up study, the incidence of dementia was 14.9% among older adults with MCI [32].

The psychological domain evaluated by the GDS-15 classified 32 older adults (19.6%) as being at moderate risk of developing subclinical or minor depression [33,34]. This clinical condition can cause functional impairment [35] and is also a risk factor for major depressive disorder and suicidal behavior [36,37], in addition to its association with functional impairment, poor quality of life, and increased health costs [38,39]. Since patients with depression do not initially present with specific symptoms [40], it is important to screen for possible disease in order to prevent the onset/deterioration of the clinical condition.

Regarding the sensory domain, 88 older adults (53.9%) reported poor or fair vision and/or hearing. Changes in vision and hearing are one of the most common clinical conditions of older adults [41] and can have disastrous consequences such as restrictions in functional ability [42,43], cognitive alterations [44,45], decreased well-being and quality of life [46,47], and changes in interpersonal relationships [48]. Screening for alterations in the sensory domain can avoid worsening of other domains, such as the cognitive and psychological domains.

The SPPB, which is used to assess the locomotion domain, is divided into three parts: static balance, dynamic balance, and lower limb strength. In this domain, 103 older adults (63.1%) were classified as moderate/high risk of having reduced locomotion. A safe gait and activities of daily living require the systems that control postural balance be able to adapt to the demands of the environment [49]. Postural instability and falls are recognized as a geriatric syndrome and their consequences include high morbidity and mortality, loss of physical function, fear of falling [50], hospitalization, institutionalization, and high costs for health services [51]. A study found that, over 20 years, Brazil has spent about 2.3 billion Reais with authorizations for hospitalization in the public health system due to falls [52]. The early identification of individuals at increased risk in the locomotion domain can be crucial to face the challenge of falls and their consequences for public health [53].

Vitality can be understood as the bodily functions devoted to metabolizing dietary intake in order to produce the amount of energy necessary for maintaining optimal homeostasis [13]. With aging, changes occur in energy expenditure and metabolism that can interfere with food intake [14]. To maintain proper function of the body, there must be a balance between energy intake and expenditure. The vitality domain evaluated by the MNA showed that 49 older adults (30%) were at moderate risk of malnutrition. There are several reasons for malnutrition/pre-malnutrition in older adults. However, initial assessment of vitality is important since this domain has been recognized to be of great importance for successful aging [2].

Screening evaluations of older adults to identify conditions/diseases is a key component of healthcare. The aim is to detect an individual’s risk of developing or having the disease and to make short-, medium- and long-term decisions in order to reduce morbidity and mortality and associated costs [54]. A study aimed at exploring the association between healthcare costs and deficits in IC domains showed that visual, locomotion and psychological changes were associated with increased healthcare costs [55]. This type of evaluation should be introduced into practice in order to guide health teams which, in turn, can refer individuals at increased risk to faster and more specialized care and those at moderate or low risk to continuous care in an attempt to prevent deficits in domains and to minimize losses (Figure 6).

Proposed in 2015 by the WHO, the concept of intrinsic capacity focuses on the maintained capabilities of the elderly and not just on losses, changing the paradigm of the concept of aging. Although there seems to be consensus around the concept of IC, there is no consensus on the tools used for its evaluation, requiring more robust approaches [56]. Researches have used retrospective data to evaluate IC domains and, for this reason, they have not always used the tools designated by the WHO a priori. There is a discussion, for example, regarding which tools would best assess the vitality domain, due to the lack of consensus on its definition [56]. Another important issue is the need to develop a standard IC score, and for that the tools must be standardized. 

Furthermore, other markers could be incorporated into the IC assessment. Known as biomarkers of aging, some are easy to apply, such as body mass index, handgrip strength, waist and hip circumference. Others would be more difficult to incorporate into the public health assessment, such as DNA-based markers, protein-based markers and their modifications, immunological markers and oxidative stress markers [57].

There is no doubt that disease/risk screening is of the utmost importance in public health [54]. Assessment alone is not sufficient for effective risk screening. Once the diagnosis of the condition is made, a service network is necessary that supports the needs of individuals who require short-term care and also of those who require medium- and long-term care (Figure 6). The idea is not only to cure (short-term care) but also to prevent installation of the problem/worsening in order to promote healthy aging.

In this sense, the practical implication of the study was to show the importance of screening, and that the tools proposed by the WHO to assess the domains of intrinsic capacity can be used, being easy to apply, with low cost and with good sensitivity/specificity.

### Strengths and Limitations

Strengths: first, we had a good sample size in our research, producing statistically significant results and representing a good portion of the population of the USFs included; second, a relatively short application time of the assessment instruments (about 45 min), considering the information that can be collected; third, the possibility of application in basic health care, where the elderly are advised to seek the first assistance and, finally, considering the low cost of application and the results collected, the study demonstrates potential, being able to be applied to a biggest population.

Limitations: first, it is worth mentioning that only one method was used to measure the vitality domain (MNA, as designated by the WHO). Vitality needs further investigation with other tools (equally easy and cheap), mainly due to the difficulty of definition. It should be considered that a cross-sectional study, although not considered a limitation, raises important issues such as not being able to determine cause and effect. For this, studies with a longitudinal design must be carried out.

## 5. Conclusions

The results permit us to conclude that the scores of the assessed CI domains were affected by the risk category. In addition, there were individuals in all risk groups in the five domains evaluated, demonstrating the importance of this type of screening as a public health strategy, making it possible to know which risk category each elderly person belongs to and thus develop strategies in the short-, medium- and long-term.

## Figures and Tables

**Figure 1 ijerph-20-04227-f001:**
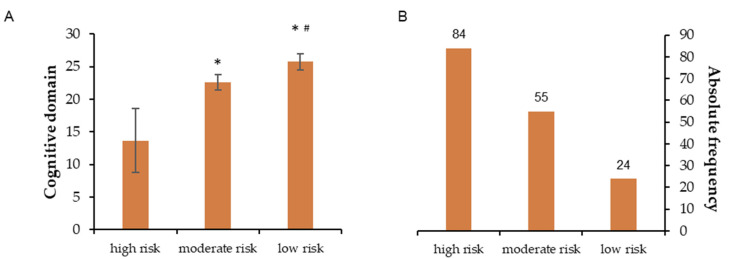
(**A**): Mean ± standard deviation for each risk group of the cognitive domain. * *p* < 0.001 compared to high risk; #: *p* = 0.002 compared to moderate risk. (**B**): Absolute frequency of older adults in each risk group.

**Figure 2 ijerph-20-04227-f002:**
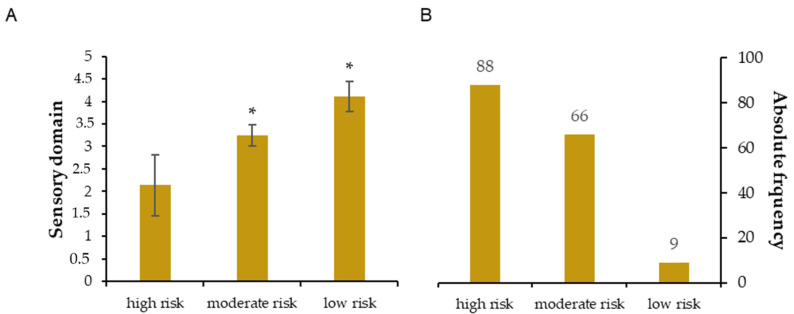
(**A**): Mean ± standard deviation for each risk group of the sensory domain. * *p* < 0.001 compared to high risk. (**B**): Absolute frequency of older adults in each risk group.

**Figure 3 ijerph-20-04227-f003:**
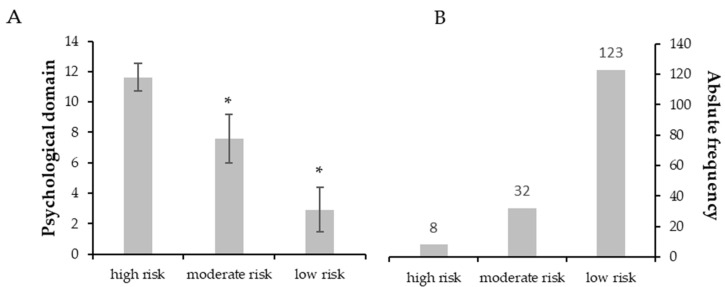
(**A**): Mean ± standard deviation for each risk group of the psychological domain. * *p* < 0.001 compared to high risk. (**B**): Absolute frequency of older adults in each risk group.

**Figure 4 ijerph-20-04227-f004:**
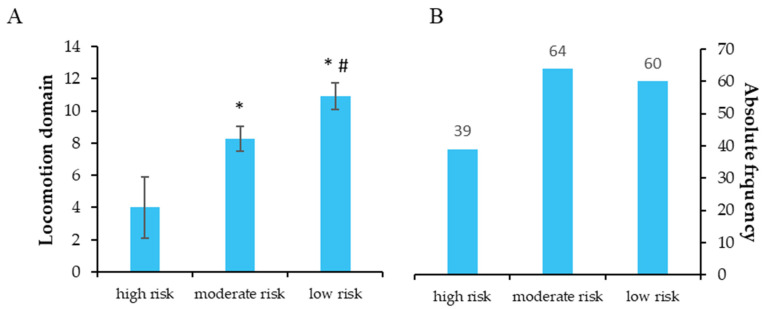
(**A**): Mean ± standard deviation for each risk group of the locomotion domain. * *p* < 0.001 compared to high risk; #: *p* = 0.002 compared to moderate risk. (**B**): Absolute frequency of older adults in each risk group.

**Figure 5 ijerph-20-04227-f005:**
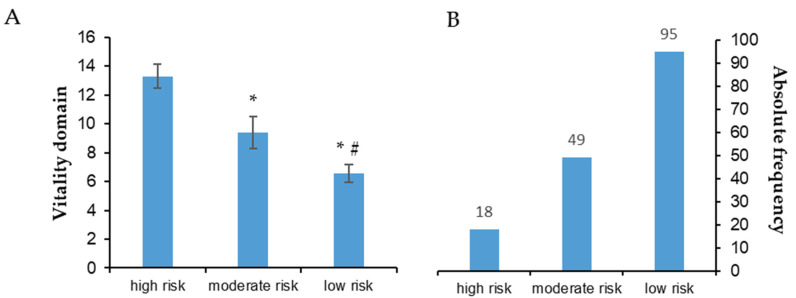
(**A**): Mean ± standard deviation for each risk group of the vitality domain. * *p* < 0.001 compared to high risk; # *p* = 0.002 compared to moderate risk. (**B**): Absolute frequency of older adults in each risk group.

**Figure 6 ijerph-20-04227-f006:**
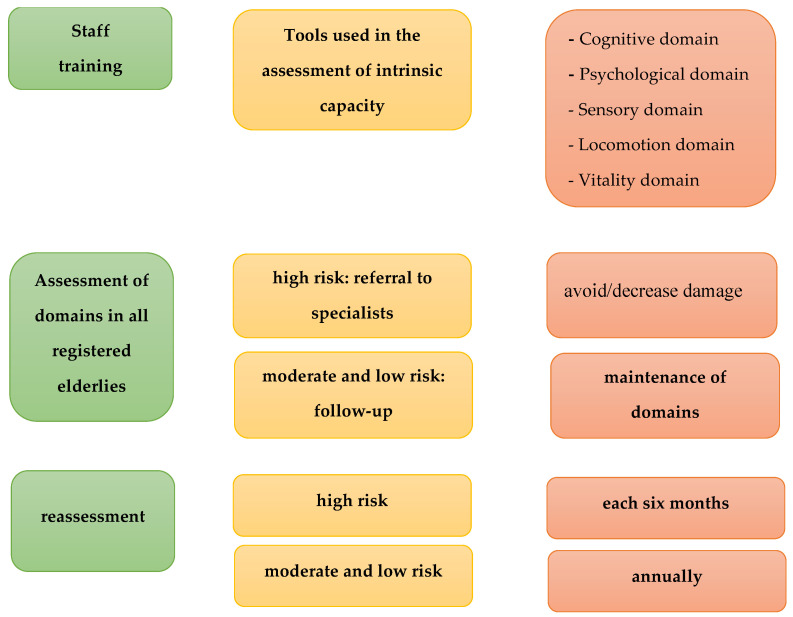
Algorithm for implementing and monitoring the domains of intrinsic capacity in public health.

**Table 1 ijerph-20-04227-t001:** Sample characteristics.

	Total	Female	Male
Participants	163	94	69
Age (years)	72.22 ± 7.60	72.18 ± 7.99	72.28 ± 7.11
Level of education (n, %)			
Illiterate	13 (7.97)	11 (11.70)	2 (2.89)
Literate	14 (8.58)	7 (7.44)	7 (10.44)
Elementary School			
Incomplete	89 (54.60)	51 (54.25)	37 (53.62)
Complete	16 (9.81)	11 (11.70)	6 (8.69)
High school			
Incomplete	7 (4.29)	2 (2.12)	5 (7.24)
Complete	12 (7.36)	7 (7.44)	5 (7.24)
Higher education			
Incomplete	1 (0.61)	0	1 (1.44)
Complete	10 (6.13)	4 (4.25)	6 (8.69)
Postgraduate	1 (0.61)	1 (1.06)	0
Living arrangements (n, %)			
Alone	26 (15.95)	20 (21.27)	6 (8.69)
Accompanied	137 (84.04)	74 (78.72)	63 (91.30)
Medications	3.59 ± 2.67	3.87 ± 2.59	3.22 ± 2.75
Habits (n, %)			
Alcohol	40 (24.53)	16 (17.02)	24 (34.78)
Smoke	75 (46.01)	30 (31.91)	45 (65.21)

## Data Availability

The datasets generated during and/or analyzed during the current study are available from the corresponding author upon reasonable request.

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
