# Peer review of "Use of Intrinsic Capacity Domains as a Screening Tool in Public Health"

_ijerph, 2023, doi:10.3390/ijerph20054227_

Round 1

Reviewer 1 Report

The screening of the multidimensional condition of the elderly is interesting. However, the cross-sectional methodological design limits the scientific contribution and resulted in confusion in the objectives, discussion and conclusion.

The results presented do not seem to be the most recommended for an adequate interpretation.

The present study makes it possible to analyse the profile of the investigated sample in terms of intrinsic screening domains.

However the obtainment of elderly in the three categories does not enable the conclusion that there is importance in screening this population.

Author Response

The authors appreciated the comments. Please see the attachment

Reviewer 2 Report

This is a concise and solid study that exemplifies the use of intrinsic capacity, as recommended by WHO, and encourages its further application in clinical screening. The topic is of practical and educational interest. A minor recommendation is to expand the discussion, with more elaboration considering the perspectives of using intrinsic capacity measurements. Clearly, the present measurements recommended by WHO may be not the most effective and informative measurements imaginable, and may be further improved. Hence, it may enrich the discussion to elaborate more on the current drawbacks of the recommended measurements, as well as potential desirable improvements from the practical standpoints. In particular, the author may elaborate more on enhancing functional measurements with additional biomarkers of aging.

Author Response

The authors appreciated the comments. Please see the attachment.

Reviewer 3 Report

Thank you for the opportunity to review your manuscript. I hope the comments are helpful.

§  The abstract could include some effect estimates. It is currently a little too descriptive.

§  The objectives and hypotheses can be more clearly articulated in the Introduction. The knowledge gaps can be elaborated further.

§  The first two paragraphs of the Materials and Methods are brief. What is the detailed inclusion and exclusion criteria of participants?

§  The table may be hard to read. There is a mixture of terms used: n and Number. The use of mean and standard deviation for continuous variables should be stated in the tables too.

§  The phrase “there is an effect on risk on the..” is unclear.

§  The figures are difficult to understand. Some bars indicate the results at the top, while bar do not have numbers at the top.

§  Unfortunately, the Results section is too brief. Much more text may be needed to better evaluate this article.

§  The impact and implications of the research work can be further elaborated in the Discussion.

§  The Discussion section also does not have a strengths and limitations section.

§  The Conclusion is brief and general too.

§  Figure 6 is hard to read. It is unclear what the single arrow that goes from the top right to bottom left means.

§  Overall, the findings need to be elaborated. More text is required to explain the tables and figures.

§  Some minor proofreading and formatting are needed.

Author Response

(The authors gave the same response as above.)
